

# Three-dimensional printing with biomaterials in craniofacial and dental tissue engineering

Wen Liao[1,2], Lin Xu[2], Kaijuan Wangrao[2], Yu Du[2], Qiuchan Xiong[2,3] and Yang Yao[2,3]

[1] Department of Orthodontics, West China Hospital of Stomatology, Sichuan University, Chengdu, Sichuan, China
[2] State Key Laboratory of Oral Diseases, West China Hospital of Stomatology, Sichuan University, Chengdu, Sichuan, China
[3] Department of Oral Implantology, West China Hospital of Stomatology, Sichuan University, Chengdu, Sichuan, China

## ABSTRACT

With the development of technology, tissue engineering (TE) has been widely applied in the medical field. In recent years, due to its accuracy and the demands of solid freeform fabrication in TE, three-dimensional printing, also known as additive manufacturing (AM), has been applied for biological scaffold fabrication in craniofacial and dental regeneration. In this review, we have compared several types of AM techniques and summarized their advantages and limitations. The range of printable materials used in craniofacial and dental tissue includes all the biomaterials. Thus, basic and clinical studies were discussed in this review to present the application of AM techniques in craniofacial and dental tissue and their advances during these years, which might provide information for further AM studies in craniofacial and dental TE.

## INTRODUCTION

The development of tissue engineering (TE) and regeneration constitutes a new platform for translational medical research. It has already been an important kind of therapeutic method in craniofacial and dental field, such as trauma, skeletal disease, wound surgery and periodontal disease (*Rai et al., 2017*). There are several approaches to develop scaffolds, such as electrospinning, mold casting, salt leaching, sintering and freeze drying. Some of these methods are easy and inexpensive, such as mold casting and salt leaching. Some can fabricate three dimensional scaffolds with good structure with a comparatively high speed, such as electrospinning, however, none of them can solve the problem of solid freeform fabrication. Solid freeform fabrication of three-dimensional scaffolds with complex space structure, not only the irregularly curved external structure, but also the internal porous structure, is important in craniofacial and dental regeneration because of its anatomical limitations. Therefore, attempts to improve design and fabrication of bio-active scaffolds, especially on freeform fabrication comprise majority of studies in biomaterial researches.

Corresponding author
Yang Yao, yaoyang9999@126.com

Recently, additive manufacturing (AM) has been applied for scaffold developing (*He et al., 2015*). This method was firstly introduced by Herver Voelcker in 1970 to describe the algorithms for the purposes of 3D solid modeling. AM has been widely used in industry because of its accuracy of shaping (*Torres et al., 2011a*; *Torres et al., 2011b*). It helps researchers to meet the demands of solid freeform fabrication in TE, too (*Warren et al., 2003*; *Obregon et al., 2015*). It also has unique advantages in fabrication of patient-specific scaffolds with multiple materials. In some recent advances, materials with live cells were used, making it possible to construct organ and tissue using AM (*Mannoor et al., 2013*).

Another hots spot of study in the field of tissue engineering combined with material manufacturing methods is electrospinning. Electrospinning uses electrostatic principle to manufacture the nanofibers required for TE applications (*Zamani et al., 2018*). There are mainly three types of technique: blending electrospinning, coaxial electrospinning, and emulsion electrospinning; they share the same basis (*Lu et al., 2016*; *Tong, Wang & Lu, 2012*). There is a high electric field applied to draw a polymer solution between the injection needle and a collector. The polymer forms a suspended drip and is stretched into a conical shape called "Taylor Cone" by the high voltage power. Then, the charged droplet forms a charged jet by breaking free from the surface tension of the top droplet. Due to the evaporation of the solvent or the curing cooling of the solute and melt, the charged jet finally condenses into filaments and deposits on the collecting plate in the form of nonwovens (*Barnes et al., 2007*; *Nair, Bhattacharyya & Laurencin, 2004*; *Chan et al., 2009*). The nanofibers prepared by electrospinning have large specific surface area and high porosity in three-dimensional structure, which makes electrospinning nanofiber membranes have a wide application value in many fields (*Qian et al., 2011*; *Chung et al., 2010*). It is worth mentioning that bio-electrospraying and cell electrospinning, both based on this principle, were firstly used to deal with living cells and whole organisms in 2005/06 (*Jayasinghe, Qureshi & Eagles, 2006*; *Townsend-Nicholson & Jayasinghe, 2006*). A series of studies have confirmed that this high-strength electric field drive technology, naming bio-electrospraying, showed no significant side effect on the bioactivity of living samples (*Jayasinghe, 2011*). Cell electrospinning is a leading technology in the formation of cell fibers and stents that can be used to create a variety of biological structures, from simple cell stents and diaphragms to more complex structures (*Jayasinghe, 2013*). In the recent years, bio-electrospraying and cell electrospinning have attracted significant increasing amount of interest.

Here we review the application of AM techniques in craniofacial and dental TE. First, we will describe the types and strategies of four typical AM printers used by tissue engineering researchers most frequently, along with their advantages and limitations. Then, we will present recent advances of AM related with craniofacial bone, craniofacial cartilages and dental tissue. Finally, we will look ahead to recommend the future possible AM research field in craniofacial and dental TE.

## SURVEY METHODOLOGY

PubMed and Web of Science databases were searched (until January 2018) using the following free-text terms: additive manufacturing, craniofacial/dental tissue engineering.

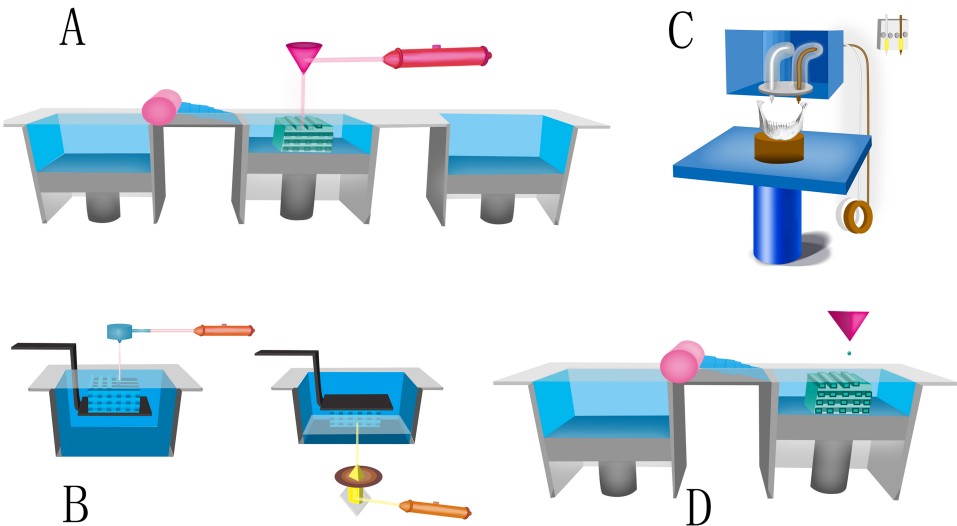

**Figure 1** **Four kinds of typical AM printers.** (A) Schematic of SLS. The fabrication chamber is settled at the base, filling with tightly compacted plastic powder. When the laser beam moves under the guidance of the scanner system and computer code, precisely shaped monolayer is printed by causing the temperature to rise above the melting point of plastic powder. (B) Schematic of SLA. A computer-controlled laser beam moves and cures the top liquid resin by photopolymerisation. The polymerized resin will adhere to a building platform for support. After finishing the first layer, the building platform drops a defined distance under the liquid surface and the laser repeats the above steps to cure a second layer. (C) Schematic of FFF. Thermoplastic polymeric filament is extruded as the "ink" from a high temperature nozzle (typically 95 °C–230 °C) because of a solid-semiliquid state transition. After printing the pattern of the first layer on a surface, either the nozzle rises, or the platform descends in the $Z$-axis direction at a thickness of a mono by the control of computer. The process is repeated until structure generation is complete. (D) Schematic of binder jetting: Liquid binder is printed as ink onto powder container. Then a new consecutive solid thin layer of free powder will be put on the binder. This printing process repeats until finishing the work.

## AM Approaches in craniofacial and dental TE
### Selective Laser Sintering (SLS)

SLS was developed by Carl Deckard at the University of Texas and described in his master's thesis (*Deckard, 1991*; *Deckard, Beaman & Darrah, 1992*; *Beaman & Deckard, 1990*). Its fundamental principle is to control the laser concentrated infrared heating beam to melt free powders together to generate a precise structure. In a SLS printer, a fabrication chamber is settled at the base, filling with tightly compacted plastic powder. The temperature of the chamber is kept just below the melting point of free powder. While the laser beam moves under the guidance of scanner system and computer code, precisely shaped monolayer is printed by causing the temperature to rise above the melting point of plastic powder (*Melchels, Feijen & Grijpma, 2010*) (Fig. 1A). As a result, morphology and melting temperature of the powder are considered as the two crucial parameters in laser sintering (*Mazzoli, 2013*). According to the mechanism of SLS, the heating temperature should be able to melt the surface layer. The molten materials on the surface then work as binder to connect neighboring non-molten particle cores (*Mazzoli, 2013*). This so-called "partial melting" phenomenon was modeled first by *Fischer et al. (2002)*. The laser sintering

powder is commercially available. They are polymeric materials such as poly(L-lactide) (PLLA) /carbonated hydroxyapatite (CHA) (*Zhou et al., 2008*), polyvinyl alcohol (PVA) (*Chua et al., 2004*) and poly-e-/caprolactone (PCL) (*Williams et al., 2005*). In a SLS printer, polymeric powder have a 50 $\mu$m mean particle size diameter (*Mazzoli, 2013*).

Many advantage of SLS method, such as accuracy, fast fabricating, low price, elective powder type, no need of supporting material, can be documented (*Mazzoli, Germani & Moriconi, 2007*). The disadvantage of SLS is that with crucial laser power and scanning speed, there is limit in the size of object fabricated with the commercially obtained machines. What's more, this method cannot fabricate scaffolds with hydrogel material (*Duan & Wang, 2011*).

### Stereolithography (SLA)

SLA printing was firstly published in 1986, in U.S. patent *Apparatus for production of three-dimensional objects by stereolithography* (*Hull, 1986*). He first exploited the spatially controlled solid transition of liquid-based resins by photopolymerization to produce complex structures layer-by-layer in SLA approach (*Skoog, Goering & Narayan, 2014*). In brief, a computer-controlled laser beam moves and cures the top liquid resin by photopolymerization. The polymerized resin adhere to a building platform for support. After finishing the first layer, the building platform drops a defined distance under the liquid surface and the above steps repeats to cure a second layer (Fig. 1B).

This technique was later modified by application of digital light projector, known as digital light processing (DLP). It enables architectures built from the bottom of the building platform. After finished the first layer, the platform raises a short distance from the liquid surface and curing procedure repeats. It looks like the structure is lift by the platform, so that the resin required is significantly reduced. Since DLP derived initially from SLA and they share close concepts, in this review, we use SLA to refer to them both. Taking advantage of the extreme accuracy of laser light, SLA printer has been largely used to build complex and precise structures. Most commercial systems have the capacity to fabricate structures with a resolution of 50 $\mu$m. On the other hand, the major limitation of SLA also lies on stereolithography, which limits choices of resins. Most of SLA resins are based on low molecular weight, multi-functional monomers for they formed highly cross-linked networks. Poly (propylene fumarate) (PPF) is the most often used polymer in the fabrication of tissue scaffolds with SLA because of its favorable biocompatibility and photo-cross linking functionality. Although only a limited selection of photocurable resins have been used in SLA, such as PPF and polyurethane (PU) (*Hung, Tseng & Hsu, 2014a*), efforts have been made to improve the features of photocurable materials for TE usage, in order to create biodegradable materials (*Skoog, Goering & Narayan, 2014*) and cell-compatible photocurable hydrogels, in the past decade.

### Fused deposition modeling (FDM)

FDM is another common AM technique, which was first used in the 1990s (*Cai, Azangwe & Shepherd, 2005*). The printing process of FDM is based on layer-by-layer deposition of thermoplastic polymers. Due to a solid-semiliquid state transition, thermoplastic polymeric filament is extruded as the "ink" from a high temperature nozzle (usually 95 °C–230 °C).

After printing the pattern of the first layer on a surface, either the nozzle rises, or the platform descends in the $Z$-axis direction at a thickness of a monolayer under the control of computer. The process is repeated until structure generation is completed (*Korpela et al., 2013*). Depending upon the polymer material and the design, the FDM printer usually prints 3D structures with a typical thickness of 100–300 μm (*Cai, Azangwe & Shepherd, 2005*) (Fig. 1C).

This technique has unique advantages because of its wide-ranged operating temperature, user friendly control system, and large number of commercial platforms. Several kinds of biodegradable materials have been used in the process, including polylactic acid (PLA), PVA, PCL, poly (D, L-lactide-co-glycolide) (PDGA) and poly (D, L-lactide) (PDLLA). Several polymers, such as PLA, PCL and PVA, are extensively utilized for their considerable biocompatibility and biodegradation. With some modification of the printer, hydrogels such as alginate, collagen, decellularized ECM, and marine products such as biogenic polyphosphate (Bio-PolyP) and biogenic silica (Bio-Silica) (*Wang et al., 2013*; *Wang et al., 2014*) can be used as well, providing possibility of loading live cells in printing progress.

However, FDM has a significant drawback, which is the lowest precision among the four methods. The minimal scale of the printing bar is about 0.1 mm (*Cai, Azangwe & Shepherd, 2005*). It is also difficult to generate micro-porous structures for bone TE without further modifications. In addition, as it is printing in an open space, external supports is needed to get rid of the collapse of structures. After finishing the printing, those supports must be removed carefully.

### Binder Jetting

Binder jetting is a technology developed at almost the same period with FDM. Its first development is in the early 1990s (*Sachs, Cima & Cornie, 1990*). In 2010, the first binder jetting machine was commercially obtained. Its basic working process shares many similarities with inkjet printing (*Meteyer et al., 2014*). In a binder jetting printer, liquid binder is printed as "ink" onto powder container. Then a new consecutive solid thin layer of free powder will be put on the binder. This printing process repeats until work finishes. The structures printed by binder jetting printers have layer thickness among 76–254 μm (*Torres et al., 2011a*; *Torres et al., 2011b*) (Fig. 1D). The advantage of this method is that binder jetting printer has various choices of printable materials: high-performance composites are used to produce tough, strong, colored, and best resolution models, elastomeric materials which give rubber-like properties or casting material which enables the creation of metal prototypes (*He et al., 2015*). Another advantage is parts can be produced with no need of supporting structure, so it is more applicable in complicated 3D structure establishment (*Gokuldoss, Kolla & Eckert, 2017*). This method has a faster printing speed than other AM methods, which can be accelerated by using multiple print heads. On the other hand, the disadvantage of this method is also clear. A lot of post-printing treatment increased the time and financial cost. The control of pore existence, size and shape is difficult because material is stacked, not melted together.

## Current status and challenges of AM applications for craniofacial bone, cartilage and dental tissues

### AM application in craniofacial bone TE

*Polymer biomaterials for craniofacial bone TE.* Fabricating maxillofacial bone scaffold is a major application of AM technology in craniofacial usage. The selection of an ideal bone graft material relies on multiple factors such as material viability, graft size, porosity, hydrophilic, biodegradability, osteoconductivity and osteoinductivity. It was first reported that synthetic polymeric materials could generate AM bone scaffolds. Many polymers are printable, for they often have proper melting ranges to fulfill the technique requirement of shaping with FDM or binder jetting. As far back as in 1996, PLA was used as AM material in computer aided design (CAD) bone generation (*Giordano et al., 1996*). After that, other polymeric scaffolds have been increasingly developed in AM techniques, such as PCL (*Williams et al., 2005*; *Lohfeld et al., 2012*; *Korpela et al., 2013*; *Van Bael et al., 2013*; *Temple et al., 2014a*), poly(lactic-co-glycolic acid) (PLGA) (*Luangphakdy et al., 2013*), poly(trimethylene carbonate) (PTMC) (*Blanquer, Sharifi & Grijpma, 2012*) and so on. As a widely used biomedical material, PLA has good biocompatibility as implants with FDA clearance. Printed PLA bars have physical properties of maximum measured tensile strength. The maximum measured tensile strength of low molecular weight PLLA (53 000) is $17.40 \pm 0.71$ MPa, while that of high molecular weight PLLA (312,000) is $15.94 \pm 1.50$ MPa (*Giordano et al., 1996*). PCL is an alternative with PLA because it does not release acid in PLA remodeling. This means it is more resistant *in vivo*. PCL also has a lower glass transition temperature and melting temperature, making it superior to PLA in certain bone grafting applications. For instance, PCL can be easily blended with other materials, including tricalcium phosphate (TCP), hydroxyapatite (HA) and bioactive glass (BAG), due to its low melting temperature (*Korpela et al., 2013*). In addition, the compressive module of PCL can be increased up to 30–40% by adding 10 wt % of BAG.

As modifications for the mechanical performances (*Duan & Wang, 2010*), polymers are also blended in defined ratios to make printable composites, such as PCL/PLGA by FDM (*Shim et al., 2014*) and PLGA/PVA by binder jetting (*Ge et al., 2009*). PVA also serves as a porogen in the printed architectures by taking advantage of its water-soluble properties. PVA-blended HA was printed by SLS to study the feasibility of composite scaffold (*Simpson et al., 2008*). SEM observations showed significant improvements in the sintering effects and to be a suitable material when processed by SLS for TE scaffolds.

*Cells and animal models used in craniofacial bone TE.* The selection of cell is important for bone TE. For orthopedic and maxillofacial researches, primary stem cells as bone marrow stromal cells (BMSC) (*Fedorovich et al., 2009*; *Rath et al., 2012*) and adipose derived stem cells (ADSC) (*Temple et al., 2014a*) are wildly applied to seed cell types. Fibroblasts are used for viability test and proliferation essay, as well as human multi-potent dental neural crest-derived progenitor cells (dNC-PCs) (*Fierz et al., 2008*). Multiple bone cell lines are applied in AM studies, including MC3T3-E1 (*Leukers et al., 2005*; *Khalyfa et al., 2007*; *Lan et al., 2009*; *Melchels, Feijen & Grijpma, 2010*; *Blanquer, Sharifi & Grijpma, 2012*), SaOS-2

(*Duan & Wang, 2010*; *Wang et al., 2013*), C3H/10T1/2 cells (*Inzana et al., 2014*) and MG-63 (*Feng et al., 2014a*; *Feng et al., 2014b*). With osteogenic induction, the attached bone cells not only exhibited cell viability around 60%–90%, but also kept potential of osteogenic differentiation which is confirmed by observing bone metabolism related RNA and protein expression, such as runt-related transcription factor 2 (RUNX2), bone morphogenetic proteins (BMPs), alkaline phosphatase (ALP) and osteonectin (ON) activity. For cells used in craniofacial bone TE, there are different advantages for different cells. Bone cell lines as MC3T3-E1, SaOS-2, c3h/10T1/2, MG-63 were often used for initial screening of biological activity of materials (*Przekora, 2019*). Since these cells are tumor-derived cell lines or immortalized osteoblast cell lines, their gene expressions are quite different from those of primary cells (*Pautke et al., 2004*). The best seed cells for craniofacial bone TE are still considered to be primary OBs because of their behavior in studying osteoconductive and osteopromotive properties (*Przekora, 2019*). The advantage of using stem cells also include testing the osteoconductive ability of printing materials (*Temple et al., 2014b*). What's more, many kinds of tissue can be the source of autologous stem cells.

Several animals had been taken in AM mandible scaffold research. Rabbits are most frequently used in the study of mandibular bone repair (*Alfotawei et al., 2014*). A protocol described the usage of three-dimensional printed scaffolds with multipotent mesenchymal stromal cell (MSCs) in mandibular reconstruction of rabbits. They used BMSC and ADSC from rabbits (*Fang et al., 2017*). One of the previous studies was performed on six mature minipigs (Fig. 2). The researchers created four mandibular defects on each pig. After the defect sites were modelled by CAD/CAM techniques, scaffolds with complex geometries and very fine structures were produced by AM technology. Then the autologous porcine bone cells were seeded on these polylactic acid/polyglycolic acid (PLA/PGA) copolymer scaffolds. Implanting these tissue-constructs into the bone defects supported bone reconstruction (*Meyer, Neunzehn & Wiesmann, 2012*). What's more, in a recent study, researchers proved that the craniofacial reconstruction including mandible could be achieved through 3D bioprinting. They presented an integrated tissue-organ printer (ITOP) that can fabricate stable, human-scale tissue constructs of any shape. They also found vascularized bone growth in the central and peripheral portion *in vivo* trails of rats (*Kang et al., 2016*). For periodontal bone regeneration, at least 4 mm augmentations of craniofacial bone had already been achieved with synthetic monetite blocks. 3D printing TCP plates were used as onlay grafts in periodontal surgery. The 4.0- and 3.0-mm high blocks were filled with newly formed bone with 35% and 41% of respective volumes (*Torres et al., 2011a*; *Torres et al., 2011b*). These 3D-printed customized synthetic onlay grafts were further used in dental implant surgery to achieve bone augments (*Tamimi et al., 2014*). Direct writing (DW) technology had been applied to produce a TCP scaffolds to repair the rabbit trephine defect. The scaffolds had micropores ranging from $250 \times 250$ μm up to $400 \times 400$ μm. After 16 weeks, 30% of the scaffold was remodeled by osteoclast activity with new bone filling in the scaffolds and across the defects (*Ricci et al., 2012*). These studies suggested that AM scaffold with tissue engineering could be used in human craniofacial defect repair in the future.

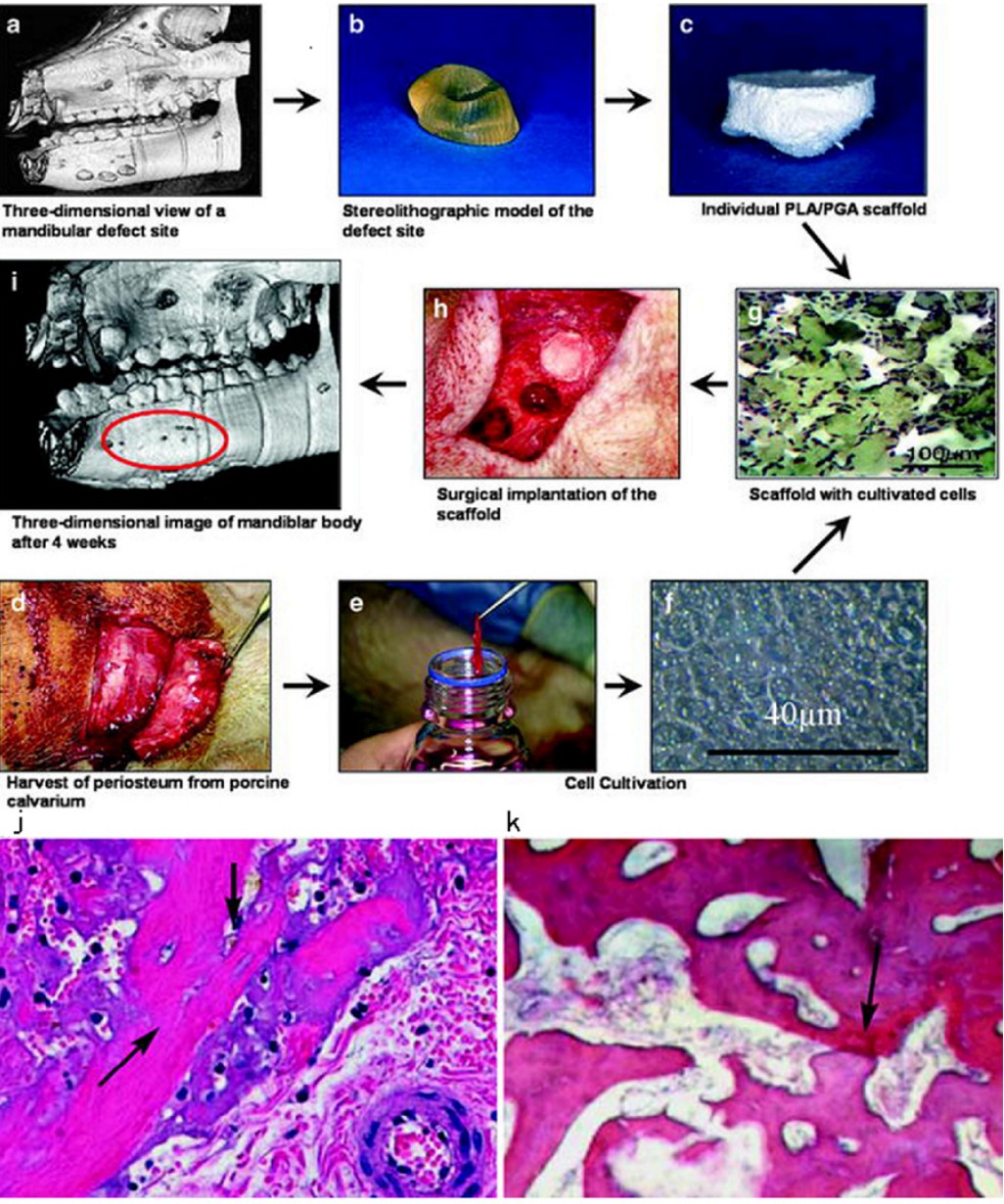

**Figure 2** **Chart of the different working steps done in this investigation.** Chart of the different working steps done in this investigation. (A–C) Fabrication of the scaffolds. (D–F) cell cultivation. (G–I) implantation of cell-loaded scaffolds and healing. Histology of bone regeneration 3 days after implantation (arrows mark regions of mineralized matrix; original magnification X10) (J). Defect site 30 days post implantation (arrows mark regions of mineralized matrix; original magnification X10) (K). © Springer (*Meyer, Neunzehn & Wiesmann, 2012*).

*Technique challenges for craniofacial bone printing and current strategies.* Although cell migration and proliferation inside the porous scaffold were observed in an AM HA scaffolds with inner-connective pores (*Fierz et al., 2008*), for all the porous scaffolds, it is still a big challenge to keep good cell viability in the central area. Insufficient nutrition and

oxygen in static culture lead to cell necrosis and make low cell density area. The method of dynamic cultivation can partly solve this problem. A dynamic cultivation system by perfusion containers strongly increased the MC3T3-E1 population compared to the static cultivation method in a 7-day *in vitro* cultivation. Close contact between cells and HA granules were observed deeply in the printed structure (*Leukers et al., 2005*). In another study, application of perfusion bioreactor system to a BCP binder jetting fabricated scaffold not only successfully reversed the decreased OB and BMSC cell numbers but also increased their differentiation potential (*Rath et al., 2012*).

Incomplete healing is another current limitation to AM bone grafts. Therefore, growth factors are applied in scaffolds. Bone morphology protein-2 (BMP2), a bone growth factor with strong bone induction property, is often used. The controlled release of BMP2 can be achieved by surface coating or nanoparticles embedding. More consideration is required according to the printing procedure for AM scaffolds. BMP2 loaded gelatin microparticles (GMPs) was used as a sustained release system and dispersed in hydrogel-based constructs, comparing with direct inclusion of BMP2 in alginate or control GMPs (*Poldervaart et al., 2013*). In another study with a multi-head deposition system (MHDS) , rhBMP2 was loaded by either gelatin (for short-term delivery within a week) or collagen (for long-term delivery up to 28 days) and dispensed directly into the hollow microchannel structure of PCL/PLGA scaffold during the printing process (*Shim et al., 2014*). The *in vivo* micro-computed tomography (micro CT) and histological analyses indicated that CL/PLGA/collagen/rhBMP2 scaffolds lead to superior bone healing quality at both 4 and 8 weeks, without inflammatory response. Transforming growth factor-β (TGF-β) was another important growth factor widely used in osteoblast differentiation and animal models (*Nikolidakis et al., 2009*).

Due to the hydrophobic feature of most printable materials, surface modification can be exploited to improve biocompatibility. Collagen is a widely used coating material for AM bone scaffold coating. The flexural strength and toughness of a calcium phosphate scaffold was significantly improved by coating a 0.5 wt% collagen film (*Inzana et al., 2014*). Biomimetic and β-TCP (*Luangphakdy et al., 2013*) can enhance the surface roughness and increase bone differentiation, thus may minimizing the need for expensive bone growth factors (*Gibbs et al., 2014*) (Table 1).

### AM application in craniofacial cartilage

*Polymer biomaterials for craniofacial cartilage TE.* Cartilage is one of the few tissues that are not vascularized, which makes its regeneration unique. The most widely applied techniques in cartilage printing included FDM, SLA and SLS. For cartilage repair, polymeric materials like PLA, PCL as well as PLGA were most common cartilage scaffolds. Another kind of major material was the hydrogel. Hydrogel could mimic the elastic module of cartilage and have been applied for cartilage reparation for a long time. Recent study showed PEG hydrogel had promising potential for cartilage bioprinting (*Cui et al., 2012*).

*Cells for craniofacial cartilage TE in AM approaches.* Chondrocytes were the standard seed cells in cartilages TE, but chondrocytes from different cartilage subtypes exhibited different

Liao et al. (2019), *PeerJ*, DOI 10.7717/peerj.7271

**Table 1  Comparison of various printed bone scaffolds in several in vitro and in vivo studies.**

| Authors | Materials | Strategies | Evidence | Model of study | Periods | Effects |
|---|---|---|---|---|---|---|
| *Leukers et al. (2005)* | HA | DP+ Sintered | In vitro | MC3T3-E1 | 7 days | The cells proliferated deep into the structure forming close contact HA granules. |
| *Williams et al. (2005)* | PCL | SLS | In vitro In vivo | BMP7 transduced HGF, Mice | 4 weeks | SLS printed PCL scaffolds enhance bone tissue in-growth. |
| *Mapili et al. (2005)* | PEGDMA | SLA | In vitro | Acryl-PEG-RGD | 24 h | Heparan sulfate allows efficient cell attachment and spatial localization of growth factors. |
| *Arcaute, Mann & Wicker (2006)* | PEGDMA | SLA | In vitro | Human dermal fi-broblasts | 24 h | Cell viability reaches at least 87% at 2 h and 24 h following fabrication. |
| *Li et al. (2007)* | epoxy resin (SL, 7560, Huntsman); CPC(scaffold) | SLA | In vitro | OB | 7 days | Negative molds were generated by SLA. Cell density increased. |
| *Khalyfa et al. (2007)* | TCP/TTCP | 3DP, Sintered, polymer infiltration | In vitro | MC3T3-E1 | 3 weeks | Objects with high compression strengths are obtained without sintering. Cell proliferation and osteogenic differentiation are achieved. |
| *Goodridge et al. (2007)* | | SLS | In vivo | Rabbit tibiae | 4 weeks | Bone was seen to have grown into the porous structure of the laser-sintered parts. |
| *Habibovic et al. (2008)* | Bioceramic | 3DP | In vivo | 12 adult Dutch milk goats | 12 weeks | Bone formation within the channels of both monetite and brushite, indicate osteoinductivity of the materials. |
| *Lee et al. (2008)* | PPF/DEF | SLA | In vitro | Fibroblasts | 1 week | Cells were adhering to and had proliferated at the top surface of the scaffold. |
| *Geffre et al. (2009)* | Polymer (NG) | FDM | In vivo | Femoral condyles (animal NG) | 5 months | Biomimetic porous design largely enhances bone ingrowth. |
| *Lan et al. (2009)* | PPF/DEF | SLA | In vitro | MC3T3-E1 | 2 weeks | MC3T3 pre-osteoblast compatibility with PPF/DEF scaffolds is greatly enhanced with biomimetic apatite coating |
| *Fedorovich et al. (2009)* | photosensitive hydrogel (Lutrol) | Hydrogel extrusion, UV | In vitro | MSCs | 3 weeks | MSCs embedded in photopolymerizable Lutrol-TP gels remain viable of 60% and keep potential of osteogenic differentiation. |
| *Zigang et al. (2009)* | PLGA/PVA | 3DP | In vitro | Human Osteoblasts CRL-11372 | 3 weeks | Expression of ALP and osteonectin remain stable whilst collagen type I and osteopontin decrease. |
| *Ge et al. (2009)* | PLGA/PVA | 3DP | In vivo | Rabbit: 1 intra-periosteum model. 2 bone defect of Ilium. | 24 weeks | In both models, the implanted scaffolds facilitated new bone tissue formation and maturation. |

Liao et al. (2019), *PeerJ*, DOI 10.7717/peerj.7271

**Table 1** (*continued*)

| Authors | Materials | Strategies | Evidence | Model of study | Periods | Effects |
|---|---|---|---|---|---|---|
| *Duan & Wang (2010)* | Customized Ca–P/PHBV | SLS | In vitro | SaOS-2, C3H10T1/2 cells | 3 weeks | Affinity of rhBMP2 on immobilized heparin facilitated the osteogenic differentiation of C3H10T1/2 cells during the whole period. |
| *Warnke et al. (2010)* | TCP, HAP | 3DP+ Sintered | In vitro | Primary human osteoblasts. | 1 week | Superior biocompatibility of HAP scaffolds to BioOss@ is proved, while BioOss@ is more compatible than TCP. |
| *Melchels, Feijen & Grijpma (2010)* | poly(D,L-lactide) resin | SLA | In vitro | MC3T3 | 11 days | Pre-osteoblasts showed good adherence to these photo-crosslinked networks. |
| *Detsch et al. (2011)* | HA, TCP, HA/TCP | 3DP | In vitro | RAW 264.7 cell line | 21 days | The results show that osteoclast-like cells were able to resorb calcium phosphate surfaces consisting of granules. |
| Torres et al. (2011) | b-TCP powder | 3DP | In vivo | Rabbit calvaria vertical bone augmentation | 8 weeks | Synthetic onlay blocks achieve vertical bone augmentations as as high as 4.0 mm. |
| *Rath et al. (2012)* | biphasic calcium phosphate (BCP) | 3DP + Sintered | In vitro | OB BMSC | 3 weeks, 6 weeks | Application of a bioreactor system increases the proliferation and differentiation potential |
| *Blanquer, Sharifi & Grijpma (2012)* | PDLLA 3-FAME/NVP | SLA | In vitro | MC3T3 | NG | Mouse preosteoblasts readily attach and spread onto porous structures with the well-defined gyroid architectures by SLA. |
| *Korpela et al. (2013)* | PCL/bioactive glass(BAG), PLA | FDM | In vitro | Fibroblasts | 2 weeks | FDM printed PLA has better cell friendly surface than PCL and PCL/BAG. |
| *Luangphakdy et al. (2013)* | PLGA TCP PPF HA TyrPC MCA | 3DP VS SLA VS PL VS CM | In vivo | Canine Femoral Multi-Defect Model | 4 weeks | TyrPCPL/TCP and PPF4SLA/HAPLGA Dip are better in biocompatibility than PLGA and PLCL scaffolds. MCA remains the best. |
| *Wang et al. (2013)* | biogenic polyphosphate (bio-polyP) and biogenic silica (bio-silica) | SFF/ indirect 3DP/ direct 3DP | In vitro | SaOS-2 cells, RAW 264.7 cells | 10 days | Bio-silica ans bio-polyP increase release of BMP2 while bio-polyP inhibits osteoclasts activity. |
| *Van Bael et al. (2013)* | PCL | SLS | In vitro | hPDCs | 2 weeks | The double protein coating increased cell metabolic activity and cell differentiation |
| *Feng et al. (2014a) Feng et al. (2014b)* | $\beta$-TCP | SLS | In vitro | MG-63 | 5 days, 4 weeks | The mechanical and biological properties of the scaffolds were improved by doping of zinc oxide (ZnO). |
| *Feng et al. (2014a) Feng et al. (2014b)* | nano-HAP | SLS(NTSS) | In vitro | MG-63 | 5 days | Cells adhered and spread well on the scaffolds. A bone-like apatite layer formed. |

Liao et al. (2019), *PeerJ*, DOI 10.7717/peerj.7271

**Table 1** (*continued*)

| Authors | Materials | Strategies | Evidence | Model of study | Periods | Effects |
|---|---|---|---|---|---|---|
| *Temple et al. (2014a)* *Temple et al. (2014b)* | PCL | FDM | In vitro | hASCs | 18 days | ASCs seeded on the PCL scaffold are successfully induced in to both vascular and osteogenic differentiation. |
| *Shim et al. (2014)* | PCL/PLGA | FDM | In vitro in vivo | hTMSCs Rabbit radius defect | 4 weeks 8 weeks | PCL/PLGA/collagen released rhBMP2 over one month in vitro, induced the osteogenic differentiation of hTMSCs in vitro and accelerated the new bone formation in the 20-mm rabbit radius defect. |
| *Inzana et al. (2014)* | Calcium phosphonate powder CPS | 3DP | In vitro In vivo | C3H/10T1/2 cells, Murine critical size femoral defect. | 9 weeks | 3D printed CPS are enhanced through alternative binder solution formulations. Tween improve the flexural strength of CPS.Implants are osteoconductive. |
| *Pati et al. (2015)* | PCL/PLGA ECM | FDM | In vitro In vivo | hTMSCs, Rat calvarial defect. | 8 weeks | The differentiation and mineralization may be augmented by combined effect of cell-laid extracellular matrix, exogenous osteogenic factors, and flow-induced shear stress |

differentiation. In AM cartilage regeneration, to generate different cartilage subtypes, chondrocytes were harvested from several kinds of cartilages. In one research, rib cartilage cells were co-cultured with adherent stromal cells in a porous PCL scaffolds fabricated by FDM, making a culture system which may have potential of clinical usage (*Cao, Ho & Teoh, 2003*). In one research, porcine articular chondrocytes were seeded in PLGA scaffold fabricated with liquid-frozen deposition manufacturing, cultured for a total of 28 days. Final results showed that cells proliferated well and secreted abundant extracellular matrix (*Yen et al., 2009*). Not only chondrocytes, but also stem cells were also applied in cartilages TE, such as MSCs and so on (*Pati et al., 2015*). Interestingly, bone marrow clots (MC) as a promising resource proved to be a highly efficient, reliable, and simple cell resource that improved the biological performance of scaffolds as well. The FDM printed PCL-HA scaffold incubated with MC exhibited significant improvements in cell proliferation and chondrogenic differentiation. This study suggested that 3D printing scaffolds, MC could provide a promising candidate for cartilage regeneration (*Yao et al., 2015*). Stem cell-based approach and chondrocyte-based approach were common choices for cartilage regenerations. The major advantage of using stem cells is that autologous transplantation can be implemented (*Walter, Ossendorff & Schildberg, 2019*). Unlike chondrocytes, autologous stem cells, such as BMSCs or ADSCs, are rich in source. Xenografts of chondrocytes is not a good choice for human cartilage repair for there are immunological reactions (*Stone et al., 1997*). It is also reported that chondrocytes lost the chondrogenic differentiation after several passages (*Von der Mark et al., 1977*; *Frohlich et al., 2007*). On the other hand, the stem cells may form fibrocartilage-like tissue in defect without grows factors (*Yoshioka et al., 2013*). Differences in depth of the defect also affect the cartilage regeneration, which should be selected according to research purposes (*Nixon et al., 2011*).

*AM application for TMJ cartilage.* The temporal mandibular joint (TMJ) disc is a heterogeneous fibrocartilaginous tissue which plays a vital role in its function. It was reported recently that researchers had developed TMJ disc scaffold with spatiotemporal delivery of connective tissue growth factor (CTGF) and transforming growth factor beta 3 (TGFβ3) which induced fibrochondrogenic differentiation of MSCs. They used layer-by-layer deposition printing technique with polycaprolactone (PCL) to fabricate the scaffold. CTGF and TGFβ3 were used as growth factors and human MSCs were used as seeding cells. After 6 weeks of cell culture, it resulted in a heterogeneous fibrocartilaginous matrix which was similar with the native TMJ disc in structure. Due to the possible effect of remaining PCL scaffold structure, the mechanical properties of the engineered TMJ discs by 6 weeks were approximated to the native properties (*Legemate et al., 2016*). *Schek et al. (2005)* used image-based design (IBD) and solid free-form (SFF) fabrication techniques to generate biphasic scaffolds. They found the growth of cartilaginous tissue and bone tissue after seeding different cells which demonstrated the possible therapy to regenerate TMJ joints (Fig. 3). In another study, researchers found that poly (glycerol sebacate) (PGS) might be potential scaffold material for TMJ disc engineering (*Hagandora et al., 2013*). Considering the complex geometries of TMJ cartilage, AM techniques have great

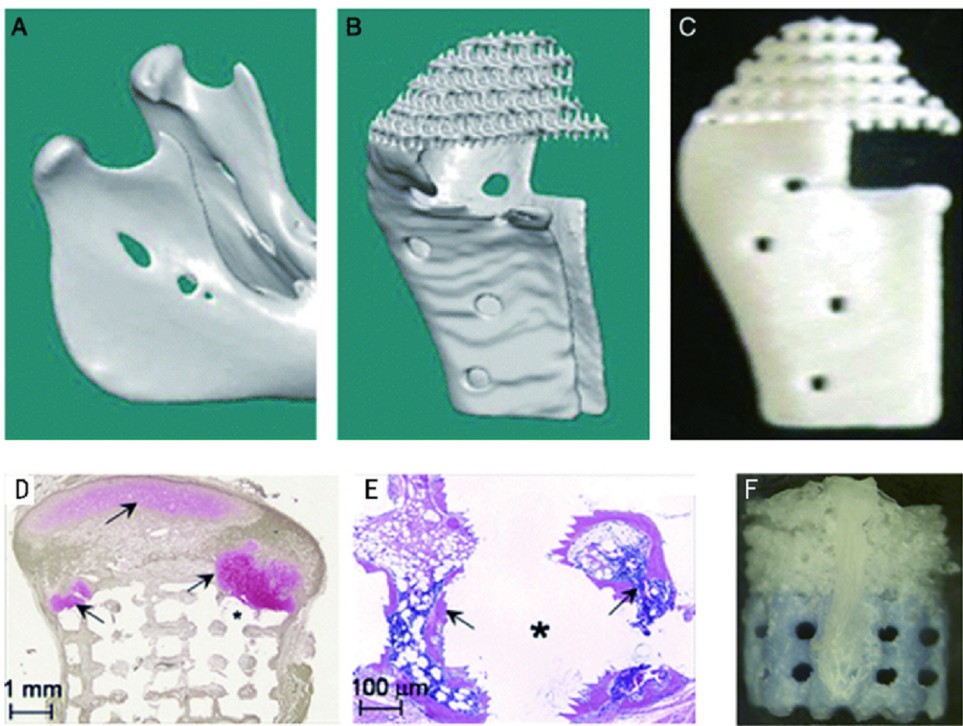

**Figure 3** **Image-based design allowing creation of defect site- specific scaffolds.** The revised legend: Image-based design allowing creation of defect site-specific scaffolds. The patient image (A) is used in conjunction with appropriate microstructure architecture to create the design for the implant (B). This design can then be produced using solid free-form fabrication, as in this prototype constructed from a single polymeric material (C). Scaffolds were demineralized prior to sectioning, resulting in empty areas (marked with *) that were previously occupied by HA. Safranin O and fast green staining showed a large area of pink-stained cartilage (arrow) in the polymer sponge, in contact with the green–brown-stained bone that formed in the ceramic phase (E). Small pockets of cartilage were also observed within the pores of the ceramic phase of the scaffold (E, arrow). Hematoxylin and eosin staining of the ceramic phase showed the formation of bone (F, arrow) with marrow space within the pores of the HA. The assembled composite: the upper polymer phase (white) and the lower ceramic phase (blue) are transversed by the two PLA struts, one of which is visible on the front of the construct (G). © John Wiley & Sons (*Schek et al., 2005*).

potential in its fabrication, and further exploration is needed in customized TMJ cartilage engineering.

*AM application for other craniofacial cartilages: ear, nose and throat.* Other than TMJ, in craniofacial area, cartilage also forms ear, nose, and larynx. Anatomically shaped ear, nose and throat were already printed through PR approaches. PCL-based ear and nose scaffold were printed and perfused with type I collagen containing chondrocytes. The samples were implanted into adult Yorkshire pigs for 8 weeks and histologically analyzed. Histological evidences present that they resulted in the growth and maintenance of cartilage-like tissue (*Zopf et al., 2015*). A bionic ear was printed with precise anatomic geometry of a human ear by alginate as matrix with 60 million chondrocytes per milliliter. An electrically conductive silver nanoparticle (AgNP) was also printed and infused inductive coil antenna as the

sensory part of the ear, connecting to cochlea-shaped electrodes supported on silicone. After *in vitro* culture, this printed bionic ear not only demonstrated good biocompatibility, but also exhibited enhanced auditory sensing for radio frequency reception, which mimicked the functional human ears (*Mannoor et al., 2013*). Functional tissue-engineering tracheal reconstruction has also been reported on rabbits by 3D printed PCL scaffolds. The shape and function of reconstructed trachea were restored successfully without any graft rejection. Histological results showed proper cartilage regeneration (*Chang et al., 2014*).

*Technique challenges for cartilage printing and current strategies.* A highlight in cartilage printing is that cells can be printed together with gels as cell vectors. For printing of cell-laden material, the important criterions lay on the suitable shear force and temperature. Otherwise, damage may occur to cells and reduce the viability in the printed constructs (*Derby, 2012*; *Pati et al., 2015*). Some studies have been paying attention to modification of the printer nozzle and materials. In one study, an electrospun head was added on an inkjet printer and print electrospun PCL film with fibrin–collagen hydrogel-based cartilage layers inside. It was designed for printing a fibrin-collagen hydrogel of five layers in only 1 mm thickness. With this multi-layer scaffold, this research successfully enhanced the strength of printed materials and overcame the major limitation of inkjet printer in material's loading ability. Therefore, it is possible to be used to print some load bearing tissue such as cartilage (*Xu et al., 2013*) (Table 2).

### AM applications in dental tissue

TE strategies for tooth and periodontal tissue regeneration have been increasingly explored recently even though the implanting of titanium artificial tooth root is clinically more and more mature (*Ohazama et al., 2004*; *Monteiro & Yelick, 2017*). By now, two tissue regeneration surgical procedures, guided bone regeneration (GBR) and guided tissue regeneration (GTR), have already been applied in dental clinics and proved to have a reliable effect on bone and gingival regeneration (*Bottino et al., 2012*). Few clinical methods can be applied in dental tissue regeneration; however, a lot of AM researches were done in this field. Multiple kinds of cells involve in the progress of dental tissue formation, including ameloblasts for enamel, odontoblasts for dentin, cementoblasts for cementum, and cells of multiple lineages including mesenchymal, fibroblastic, vascular, and neural cells that form dental pulp (*Fisher, Dean & Mikos, 2002*; *Xue et al., 2013*; *Park et al., 2014a*; *Jensen et al., 2014*). Dental tissue includes composites of enamel, dentin and pulp, periodontal ligament, cementum, and so on. Since the dental tissue are related with each other, some researches chose to establish combined dental tissue like scaffolds with AM technology, such as cementum/dentin interface (*Lee et al., 2014*) or cementum/PDL interface (*Cho et al., 2016*). Various materials can be used in AM technology for dental tissue (Table 3). As a result, we divide the load of press into one (single) tissue regeneration and multi (combined) tissue regeneration and reviewed them one by one.

*Single dental tissue regeneration.* Lee et al.'s (*2014*) group has done tooth and periodontal regeneration by cell homing. The research starts from bioprinting of PCL-HA material into two kinds of anatomically tooth shaped scaffold by SLA technology, one is human

Liao et al. (2019), *PeerJ*, DOI 10.7717/peerj.7271

*Peerj*

**Table 2  Comparison of various printed cartilage scaffolds in several in vitro and in vivo studies.**

| Authors | Materials | Strategies | Evidence | Model of study | Periods | Effects |
|---|---|---|---|---|---|---|
| *Cao, Ho & Teoh (2003)* | PCL (NaOH treated) | FDM | In vitro | hOB(iliac crest) hChondrocytes (rib cartilage) | 50 days | Osteogenic and chondrogenic cells can grow, proliferate, distribute, and produce extracellu-lar matrix in these PCL scaffolds. |
| *Smith et al. (2007)* | PCL | SLS | In vivo | Yucatan minipig mandibles | 3 months | Cartilaginous tissue regeneration along the articulating surface with exuberant osseous tissue formation. |
| *Yen et al. (2009)* | PLGA (type II collagen) | FDM | In vitro | Chondrocytes (condyles of Yorkshire pigs) | 4 weeks | Scaffolds swell slightly. The cartilaginous tissue formation was observed around but not yet in the interior of the constructs. |
| *Yen et al. (2009)* | PLGA (lyophilized for 48 h) | LFDM | In vitro | Chondrocytes (condyles of Yorkshire pigs) | 4 weeks | Decrease swelling significantly. Mechanical strength is closer to native articular cartilage. Proliferate well and secret abundant ECM. |
| *Soman et al. (2012)* | ZPR PEG | SLA | In vitro | hMSCs | 1 week | Zero Poisson's ratio (ZPR) material PEG has been printed to generate 3D printed scaffolds. The hMSCs adhere and proliferate well. |
| *Grogan et al. (2013)* | GelMA | SLA | In vitro Ex vivo | human avascular zone meniscus cells; Human meniscus ex vivo repair model | 6 weeks | Micropatterned GelMA scaffolds are non-toxic, produce organized cellular alignment, and promote meniscus-like tissue formation. |
| *Mannoor et al. (2013)* | Alginate, silicon, (AgNP infused) | syringe extrusion | In vitro | Chondrocytes (articular cartilage of calves) | 10 weeks | The ears are cultured in vitro for 10 weeks. Audio signals are received by the bionic ears. |
| *Lee et al. (2013)* | PCL, hyaluronic acid, gelatin | SLS | In vitro | Chondrocytes (New Zealand white rabbit) | 4 weeks | This study successfully forms a soft/hard bi-phase scaffold, which offers a better environment for producing more proteins. |
| *Xu et al. (2013)* | PCL, FN, Collagen | Inkjet, Electrospun | In vitro In vivo | Rabbit elastic chondrocytes; Immunodeficient mice subcutaneous model | 8 weeks | The hybrid electrospinning/inkjet printing technique simplifies production of complex tissues. |
| *Schuller-Ravoo et al. (2013)* | PTMC | SLA | In vitro | Bovine chondrocytes | 6 weeks | The compression moduli of the constructed cartilage increases 50% to approximately 100 kPa. |

**Table 2** (*continued*)

| Authors | Materials | Strategies | Evidence | Model of study | Periods | Effects |
|---------|-----------|-----------|----------|----------------|---------|---------|
| *Gao et al. (2014)* | PEG | Inkjet, UV | In vitro | human chondrocytes | 4 weeks | Printed neocartilage demonstrated excellent glycosaminoglycan (GAG) and collagen II production with consistented gene expression. |
| *Pati et al. (2015)* | dECM, PCL | Extrusion, FDM | In vitro | hASCs hTMSCs | 2 weeks | Tissue-specific dECM bioinks achieve high cell viability and functionality. |
| *Chen et al. (2014)* | PCL (coating with collagne) | SLS | In vivo | Subdermally dorsal model of female nude mice | 8 week | Collagen as a surface modification material is superior to gelatin in supporting cells growth and stimulating ECM protein secretion. |
| *Chang et al. (2014)* | PCL | FDM | In vivo | Rabbit half-pipe-shaped tracheal defect. Rabbit MSCs | 8 weeks | The 3DP scaffold with fibrin/MSCs served as a resorbable, chondro-productive, and proper cartilage regeneration strategy. |
| *Zhang et al. (2014)* | PEG/ $\beta$-TCP | SLA & hydrogel | In vivo | Rabbit trochlea critical size osteochondral defects. | 52 weeks | The repaired subchondral bone formed from 16 to 52 weeks in a "flow like" manner from surrounding bone to the defect center gradually. |
| *Yao et al. (2015)* | PCL/HA | FDM | in vitro in vivo | Bone marrow clots and BMSC from 30 female New Zealand white rabbits (5-6 months old). 60 Female nude mice (6-7 weeks old). | 4 weeks | Combination with MC is a highly efficient, reliable, and simple method that improves the biological performance of 3D PCL/HA scaffold. |
| *Zopf et al. (2015)* | PCL | SLA | In vitro In vivo | Yorkshire pigs Supraperichondrial soft tissue flaps | 2 months | The histological evidence present that anatomically PCL based ear and nose resulted in the growth and maintenance of cartilage-like tissue. |

**Table 3  Comparison of various printed dental scaffolds in several in vitro and in vivo studies.**

| Authors | Materials | Strategies | Evidence | Model of study | Periods | Effects |
|---|---|---|---|---|---|---|
| *Kim et al. (2010)* | PCL/HA (Infused SDF1- and BMP7-loaded collagen) | FDM | In vivo | 22 male (12-week-old) Sprague-Dawley rats: 1 Rat's dorsum subcutaneous pouches for human mandibular molar scaffolds, 2 right mandibular central incisor for rat central incisor teeth | 9 weeks | A putative periodontal ligament and new bone regenerate at the interface of rat incisor scaffold with native alveolar bone by cell homing. |
| *Lee et al. (2014)* | PCL/HA 100 um, 300 um, 600 um. | FDM | In vitro In vivo | 1 DPSCs, 2 PDLSCs, 3 ABSCs. The dorsum's mid-sagittal plane for 10-week-old immunodeficient mice (Harlan) | 4 weeks | DPSC-seeded multiphase scaffolds yield aligned PDL-like collagen fibers. The fibers inserted into bone sialoprotein-positive bone-like tissue and putative cementum matrix protein 1-positive/dentin sialophosphoprotein-positive dentin/cementum tissues. |
| *Xue et al. (2013)* | Alginate/ gelatin | Hydrogel extrusion | In vitro | hDPCs | | Self-defined shaped 3D constructs are printed and achieve the cell viability of 87%. |
| *Jensen et al. (2014)* | PCL | FDM | In vitro | hDPCs | S3 weeks | The HT-PCL scaffold promotes cell migration and osteogenic differentiation. |
| *Rasperini et al. (2015)* | PCL | SLS | In vivo | Clinical case on a periodontitis patient's canine. | 13 months | The case demonstrated a 3-mm gain of clinical attachment and partial root coverage. However, the scaffold became exposed at the 13th month. |
| *Cho et al. (2016)* | PCL, collagen I gel | FDM | Ex vivo | PDLSCs seeded PCL was placed on tooth root surface defect. | 6 weeks | The new mineralized tissue layer seen in BMP-7 treated samples expressed cementum protein 1 (CEMP1) |
| *Jung, Lee & Cho (2016)* | PEG, PCL, cell-laden Alginate | Hydrogel extrusion and FDM | In vitro | | | Multiple-layer bioprinting teeth was fabricated with a frame, two kinds of cell-laden hydrogel and a support. |

molar scaffold, and another is rat incisor scaffold. Growth factors of bone morphogenetic protein-7 (BMP7) and stromal cell-derived factor-1 (SDF1) were added into the scaffold to active cell homing in vivo. These two scaffolds were orthotopically and ectopically implanted into mandibular incisor extraction socket and dorsum subcutaneous pouches of rats. After 9 weeks, tooth-like structures and periodontal integration were successfully generated by their study with endogenous cell homing and angiogenesis (*Kim et al., 2010*). High survival rates were reported in a self-defined shape engineered pulp, which was as high as 87% ± 2%. This research was done to establish a dental pulp like tissue with human dental pulp cells (hDPCs) in sodium alginate/gelatin hydrosol (8:2), and an amount of $1 \times 10^6$ cells/ml were seeded (*Xue et al., 2013*). In a recent study to generate artificial periodontal ligament (PDL) tissue, human PDL cells were seeded on anatomically FDM printing PCL/HA scaffolds. In periodontal osseous fenestration defects on nude mice, guided fiber alignment was later observed oblique orientation to the root surface 6 weeks post implant, which mimics the mature PDL fiber aliment (*Park et al., 2014b*). Another study invested the osteogenic potential of human dental pulp stem cells (hDPSCs) on different porous PCL printing scaffolds. This research used a specially designed double-layer scaffold system for better osteogenic differentiation. The first layer was nanostructured porous PCL (NSP-PCL) scaffold, and the second layer was PCL coating with a mixture of hyaluronic acid and beta-TCP (HT-PCL) scaffold. With 21 days of *in vitro* cultivation, the NSP-PCL and HT-PCL scaffolds promoted osteogenic differentiation and $Ca^{2+}$ deposition, showing promising application periodontal tissue regeneration (*Jensen et al., 2014*). A very recent clinic case first showed the SLS printed PCL scaffolds' application on a periodontal tissue regeneration in a periodontitis patient. The case demonstrated a 3 mm gain of clinical attachment and partial root coverage. However, the scaffold became exposed at the 13th month and been removed. However, it showed huge potential of AM applications for dental tissues (*Rasperini et al., 2015*) (Fig. 4).

*Combined dental tissue regeneration. Lee et al. (2014)* established a multiphase scaffold mimicking cementum/dentin interface, PDL and alveolar bone by 3D printing blended polycarprolactione/hydroxyapatite (90:10) materials. By adding adequate growth factor and culturing cells, they established PDL-like tissue, the fiber of which connects from one side dentin/cementum tissue to another side bone-like tissue, which is just similar to living PDL's anatomical property (*Lee et al., 2014*). Another recent 3D bioprinting research showed BMP7 was benefitional for cementum formation. This research established an interface between cementum and human PDL like tissue, which is novel in combining natural tissue with artificial AM tissue *in vitro*. The AM scaffold was fabricated with PLGA, and then seeded human PDLSCs. After 6 weeks of culturing, they found that cementum-like layer can be successfully formed in this interface between cementum and human PDL like tissue. They also found that BMP7 helped in cementum matrix protein 1 secretion *in vitro*, which may be good for cementum tissue establishment (*Cho et al., 2016*).

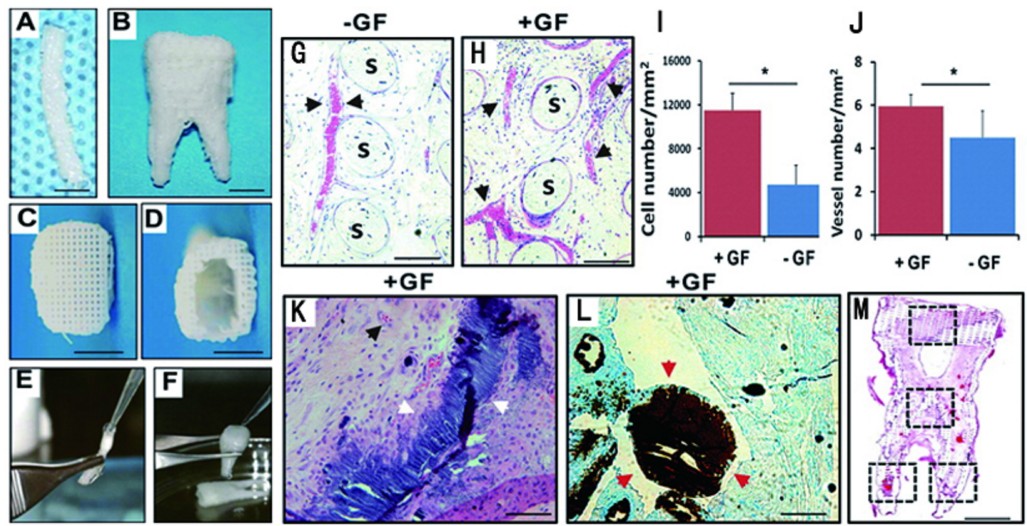

**Figure 4  Design and fabrication of anatomically shaped human and rat tooth scaffolds by 3D bioprinting.** Design and fabrication of anatomically shaped human and rat tooth scaffolds by 3D bioprinting. Anatomic shape of the rat mandibular central incisor (A) and human mandibular first molar (B) were used for 3D reconstruction and bioprinting of a hybrid scaffold of poly-$\epsilon$-caprolactone and hydroxyapatite, with 200-$\mu$m microstrands and interconnecting microchannels (diam., 200 $\mu$m), which serve as conduits for cell homing and angiogenesis (C, D). A blended cocktail of stromal-derived factor-1 (100 ng/mL) and bone morphogenetic protein-7 (100 ng/mL) was delivered in 2 mg/mL neutralized type I collagen solution and infused in scaffold microchannels for rat incisor scaffold (E) and human molar scaffold (F), followed by gelation. (G) In human mandibular molar scaffolds, cells populated scaffold microchannels without growthfactor delivery. (H) Combined SDF1 and BMP7 delivery induced substantial cell homing into microchannels. (I) Combined SDF1 and BMP7 delivery homed significantly more cells into the microchannels than without growth-factor delivery ($p < 0.01$; N = 11). (J) Combined SDF1 and BMP7 delivery elaborated significantly more blood vessels than without growth-factor delivery ($p < 0.05$; N = 11). (K, L) Mineral tissue in isolated areas in microchannels adjacent to blood vessels and abundant cells, and confirmed by von Kossa staining. (M) Tissue sections from coronal, middle, and two root portions of human molar scaffolds were quantified for cell density and angiogenesis. s, scaffold; GF, growth factor(s). Scale: 100 $\mu$m. © SAGE Publications *Kim et al. (2010)*.

# CONCLUSIONS

The transition of new techniques from a novel experimental phase to be regularly available to any laboratory has frequently driven step-changes in the progress of science (*Hung, Tseng & Hsu, 2014b*). Considering the rapid development of commercial printers and open-resource software, the AM technique has great potential to facilitate the next generation TE. Despite some limitations on current AM scaffolds, there have been recent exciting advances in AM technique microstructure control, porosity, porous interconnectivity, and surface modification, bioactivity *in vitro* and *in vivo*. Its development may lead to a promising future to functional tissue and organ regeneration. The following fields are recommended for further AM studies in craniofacial and dental TE:

The long-term healing effects on animal models.

Pre-clinic studies and clinical application on patients, including the whole procedure from the collection of defect image data of patients to the long-term morphological and functional evaluation of the AM conducted patient-specific scaffolds.

All-in-one manufacturer protocol for printing complex tissue structures with customized materials, porosity, surfaces and pattern designs.

Tissue and (or) organ printing with live cells.

### Funding

This work was supported by the National Natural Science Foundation of China (No. 81700941 and No. 31600752) and the Sichuan University-Luzhou City cooperation project (No. 2018CDLZ-14) The funders had no role in study design, data collection and analysis, decision to publish, or preparation of the manuscript.

### Grant Disclosures

The following grant information was disclosed by the authors:
National Natural Science Foundation of China: 81700941, 31600752.
Sichuan University-Luzhou City cooperation project: 2018CDLZ-14.

### Competing Interests

The authors declare there are no competing interests.

### Author Contributions

- Wen Liao conceived and designed the experiments, prepared figures and/or tables.
- Lin Xu performed the experiments, approved the final draft.
- Kaijuan Wangrao performed the experiments.
- Yu Du and Qiuchan Xiong analyzed the data.
- Yang Yao contributed reagents/materials/analysis tools, authored or reviewed drafts of the paper.

### Data Availability

The research in this article did not generate any data or code; this is a literature review.

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
