# Peer review of "Three-dimensional printing with biomaterials in craniofacial and dental tissue engineering"

_PeerJ, doi:10.7717/peerj.7271_

## Round 0.1 · original submission · Major Revisions

The reviewers would like to see some revisions made to your manuscript. I invite you to respond to the reviewers' comments and revise your manuscript. I hope that you find the comments of the reviewers useful.

·

Basic reporting

The abstract is well written.
The introduction is well written.
The survey methodology section is well written.
The body text/results/discussion sections are clear.
The conclusions are fine.

Experimental design

Not applicable, this is a review.

Validity of the findings

Not applicable, this is a review.

Additional comments

There are a number of cases where there is no space before the reference, see: "disease(Rai et al., 2017).", "developing(He et al., 2015)", "shaping(Torres et al., 2011)"; or indeed figures: "layer(Fig.1B)"

The authors should use the correct symbol for micro instead of the u, e.g.: "50um". Please correct ALL instances.

Please change "The control of porosity’s" to read "The control of porosity"

The authors correctly state "SLS was developed by Carl Deckard" but fail to cite the thesis - please correct this. Also include other research output from Deckard and coworkers.

The authors correctly state "SLA printing dates back to 1986, when Charles Hull, ‘father of 3D print’, first promoted the 60 term ‘stereolithography’ in his U.S. patent, Apparatus for production of three-dimensional 61 objects by stereolithography" but fail to cite the patent - please correct this. Also include other research output from Hull and coworkers.

The authors cortectly state "Binder jetting is a technology developed at almost the same period with FDM. Its first development is in the early 1990s. In 2010, the first binder jetting machine was commercially obtained." but fail to cite the patents/papers - please correct this.

Figure 1 legend needs to explain what A-D are.

Figure 2: legend needs correcting as it appears the image is reproduced from another journal, so the authors need the copyright permission from the publishers of those images (supply this to PeerJ with the revised manuscript), and to ensure the legend is the same as in the original manuscript, and to cite the manuscript.

Reviewer 2 ·

Basic reporting

the article is a nice article but does not do very much to the literature as there are many such review articles. also the article hasn't spoken about cell electrospining and bio-electrospraying.

Experimental design

the aims are very mundane as there are many such review articles.

Validity of the findings

the review should also talk about cell electrospinning and bio-electrospraying

Additional comments

the review is incomplete as all techniques are not discussed.

Reviewer 3 ·

Basic reporting

no comment

Experimental design

1. How many articles were searched for by the keywords described? Does table1 list all the articles and How did authors filter these articles?
2. For the articles searched, how did evaluate the authenticity and reliability of the results?
3. could the authors describe the advantages and disadvantages of different cell lines and different animal models?

Validity of the findings

The article provides a comprehensive and detailed reference about AM techniques, and evokes thinking.

Additional comments

The pictures lacked necessary legends, such as fig1, please mark the main part of AM printers.

Reviewer 4 ·

Basic reporting

The manuscript by Liao and colleagues is interesting and well presented. The topic is relevant to the field of tissue engineering and can be of interest for the readership of Peer J. The introduction well matches with the subject presentation.

Experimental design

The study is well designed and the methodology is consistent as well as the bibliography section. Figures are properly organized in order to be easy to understand and representative for the readers. The overall organization of the work is adequate.

Validity of the findings

Each work described is analyzed considering its strenghts and weaknesses.

Additional comments

The manuscript by Liao and colleagues is valuable.
Some minor revisions concerning the language has to be addressed prior to publication:
- there are mistakes in lines 135, 161, 186, 188, 298,... In general some careless mistakes can be found throughout the manuscript, please double check the work and fix all of them

- TMJ acronym has to be defined in extenso at least once

- on line 273 esophagus is wrongly reported as an organ containing cartilages!! If the authors intend cricoid cartilage close which the esophagus starts they should consider that cricoid is anatomically included in larynx.

---

## Round 0.2 · Minor Revisions

I hope the author can seize the time to solve the minor issues in the manuscript.

·

Basic reporting

The abstract is well written.
The introduction is well written.
The survey methodology section is well written.
The body text/results/discussion sections are clear.
The conclusions are fine.

Experimental design

Not applicable, this is a review.

Validity of the findings

Not applicable, this is a review.

Additional comments

The authors have revised the manuscript in line with the reviewers suggestions and the manuscript is improved.

Reviewer 2 ·

Basic reporting

the article is a good article but has ignored the work most important to this article. therefore it should discuss the work of bio-electrospraying and cell electro spinning. the following article should also be cited,

Small, 2(2006)216-219.
Analyst, 136(2011)878-890.
Biomacromolecules, 7(2006)3364-3369.
Analyst, 138(2013)2215-2223.

Experimental design

the study seems sound

Validity of the findings

the results are valid

Additional comments

the additional information meaning the introduction should contain the works citing the bio-electrospraying and cell electrospimnong.

Reviewer 3 ·

Basic reporting

no comment

Experimental design

no comment

Validity of the findings

no comment

Additional comments

This manuscript currently meets the publishing criterions and agrees to publish.

Reviewer 4 ·

Basic reporting

The manuscript has been properly integrated according to reviewers' comments and it is now suitable for publication in PeerJ. I would like to suggest the Authors to double check again the whole work since some careless mistakes can still be found.

Experimental design

see above

Validity of the findings

see above

Additional comments

see above

---

## Round 0.3 · accepted · Accept

This manuscript has been well improved. I think it's ready for publication.